# Evaluation on the Sustainability of Urban Public Pension System in China

## Qing Zhao [1]  and Haijie Mi [2,*]

1  Center for Social Security Studies, Wuhan University, Wuhan 430072, China; zhao.qing@whu.edu.cn
2  Chinese Academy of Labor and Social Security, Beijing 10029, China
*  Correspondence: mihaijie@calss.net.cn; Tel.: +86-010-6494-1017

**Abstract:** Against the background of population aging and economic downturn, the sustainability of pension systems has aroused great concern for governments across the world. To better reflect the pressure of pension payments in the changing context, the paper aims to forecast the annual pension gap of the public pension system for urban employees in China. By the use of Cohort-component population projections and stochastic projection models, the distribution of flow-based annual pension gap in the next fifty years are estimated under basic assumptions. The results show that the pension gap continues to exist from 2017 and keeps expanding until 2070 without any policy reform. Sensitivity analyses of demographics and various combinations of policy parameters on the distribution of future pension gaps are displayed. Wider pension coverage with lower policy threshold is more likely to face larger long-term pension gap.

**Keywords:** pension sustainability; annual pension gap; public pension system for urban employees; stochastic modelling; sensitivity analysis

---

## 1. Introduction

Nowadays the challenge of the sustainability of pension system has become the key point in the reform of social security system for governments all over the world. With the development of practices in various countries, the connotation of pension sustainability has been enriched and developed. In 1994, the financial sustainability of public pension system was firstly mentioned in the World Bank's report [1], in which the balance of the pension system was also measured under the influence of population aging. Holzman and Hinz [2] point out that the "sustainability" of the pension system refers to the financial stability that pension plans should have at present and in the future. It is related to the total economic output of a country. Under the constraint of the total economic output, the sustainable pension system should be able to provide the beneficiaries with the pre-promised benefits without taking any improper measures. By analyzing the effects of pension reforms, Barr [3] holds that the major concern needed to be addressed in designing the pension system is the conflict between long-term sustainable demands and the impact brought by short-term political pressure and economic fluctuation, which emphasizes on a balance between political sustainability and economic sustainability. Political sustainability depends on the level of pension benefits, pensionable age and people's choices of risk diversification. Economic sustainability depends on political support for funding to meet the demand of future pension benefits and the political feasibility in adjusting the amount of benefits. From a holistic approach, Aaron argues that the sustainable reform of pension system needs to take into account the adequacy of pension benefits and fiscal affordability, so as to achieve the goals of poverty alleviation and income smoothing to the greatest extent within institutional constraints [4,5].

With regard to the measurement of pension sustainability, there are three frequently used indicators: implicit pension debt, transitional cost and pension gap. The implicit pension debt refers to the present value of pension rights and interests accumulated by retirees and incumbents at the time point of assessment; whereas the transitional cost is the present value of total pension rights and interests accumulated by the insured population before the system transition [6]. Both implicit pension debt and transitional cost are stock-based concepts. The implicit pension debt is commonly existed among all major economies [7]. In terms of China's urban public pension system, the implicit pension debts are estimated based on different assumptions and methods. The basic conclusion could be that the problem of implicit pension debt will become more and more serious in the future [8–11]. Moreover, China is one of the few countries that have not clearly defined the scale of transitional costs and the way they are handled. The transitional cost is estimated to reach trillions of yuan if it continues to grow without any repayment [12,13]. Nevertheless, it is not enough to solely focus on implicit pension debt and transitional cost because the pressure for governments and individuals caused by the pension payment might be underestimated in a changing environment. From the perspective of financial balance, it is more important to understand the flow-based annual pension gap. The Cash-flow model is commonly used to calculate pension gap. According to the different assignment methods of input variables, cash-flow models are generally divided into three types: deterministic model, stochastic model and micro-simulation model. Actuarial agencies in the United States and Canada adopt stochastic models to make prediction about the financial situation of pension funds in the long run while UK, France, Sweden, Japan and South Korea use deterministic actuarial prediction models in their forecasts. It is widely known that the United States is at the forefront of the world in the use of stochastic forecasting model in projecting balance of social security funds. Since 1940, the Office of the Chief Actuary (OCA) affiliated to U.S. Social Security Administration has made annual forecasts of the financial revenue and expenditure of the Social Security Fund over the next 75 years and served as a measure of the sustainable development of the social security system and a basis for fund risk management. Before 2002, the prediction model used by the Social Security Administration was a deterministic one, within which the input variables of the model were a series of predetermined values, thus the output was a point estimate. If situational tests are added on this basis and input variables are assigned separately in several possible scenarios, the predictions will be generated in different scenarios. The US financial reports usually adopt three scenarios: high, medium and low. The medium one is usually the baseline scenario. Since 2003, the OCA has introduced the stochastic prediction model (OCA Stochastic Model, OSM for short) that estimates the probability distribution of future financial variables by considering the random fluctuation of one or more input variables [14]. To be specific, the stochastic prediction model is based on the historical data of each input variable: first, the time series model is constructed, then the Monte Carlo simulation method is used to carry out stochastic simulation and finally the probability distribution of pension gap is obtained [15]. Meanwhile, the Congressional Budget Office has built a long-term micro-simulation model CBOLT of social security funds based on individual longitudinal historical data since 2002 [16]. However, CBOLT model sets high requirement for data quality, therefore the deterministic model and OSM model are more frequently applied in governments' practices and researchers' studies in other countries.

Regarding the pension gap of Chinese system, previous research mostly adopts deterministic prediction cash-flow model based on sub-models of demographics, economics and pension systems [17–19], whereas very few build stochastic prediction model [20,21]. In the calculation of annual revenue and expenditure of pension systems, various results are presented based on different actuarial model settings and different assumptions of demographic, economic and policy parameters but the basic conclusions are similar: the gap will inevitably exist and remain huge in the long term without the policy adjustment and systematic reform. However, most of the estimates adopt the contribution rates for employees in formal sectors, ignoring the differentiated contribution rates for non-standard employees in informal sectors [22,23] and neglecting the fact that the actual collection

rate is less than 100% due to suspension of contribution and so forth. [24,25]. These factors are all likely to lead to an overestimation of pension revenues. In addition, some studies fail to take into account the fact of mixed-management of social pooling and individual accounts and policy intervention in pension benefits in China [26].

In view of the above mentioned, this research makes minor improvements of the basic stochastic prediction model according to the actual situation of China's pension system (see Figure 1) and includes sensitivity analysis of demographics and policy parameters, so as to make a more realistic forecast of annual pension gap and to further explore the institutional factors affecting the sustainability of public pension system in China. Especially under the background of the reform on collection and management system of social insurances and the downward adjustment of contribution rates, it is of great significance to examine the impact of changes in policy parameters on future pension gaps. The paper is organized as follows. The first section introduces the motivation of this research by reviewing current literatures on pension sustainability and pension gap models. In the second section, we review the development of China's public pension system, introduce the characteristics of public pension system for urban employees and analyze the problems of the system in terms of participation and pension fund. The third part makes basic assumptions for future pension fund and clarifies data sources. Section four presents the results under baseline hypothesis by establishing forecasting models of population and pension gap. The fifth section is sensitivity analysis on demographics and various policy scenarios. The last section is for conclusion.

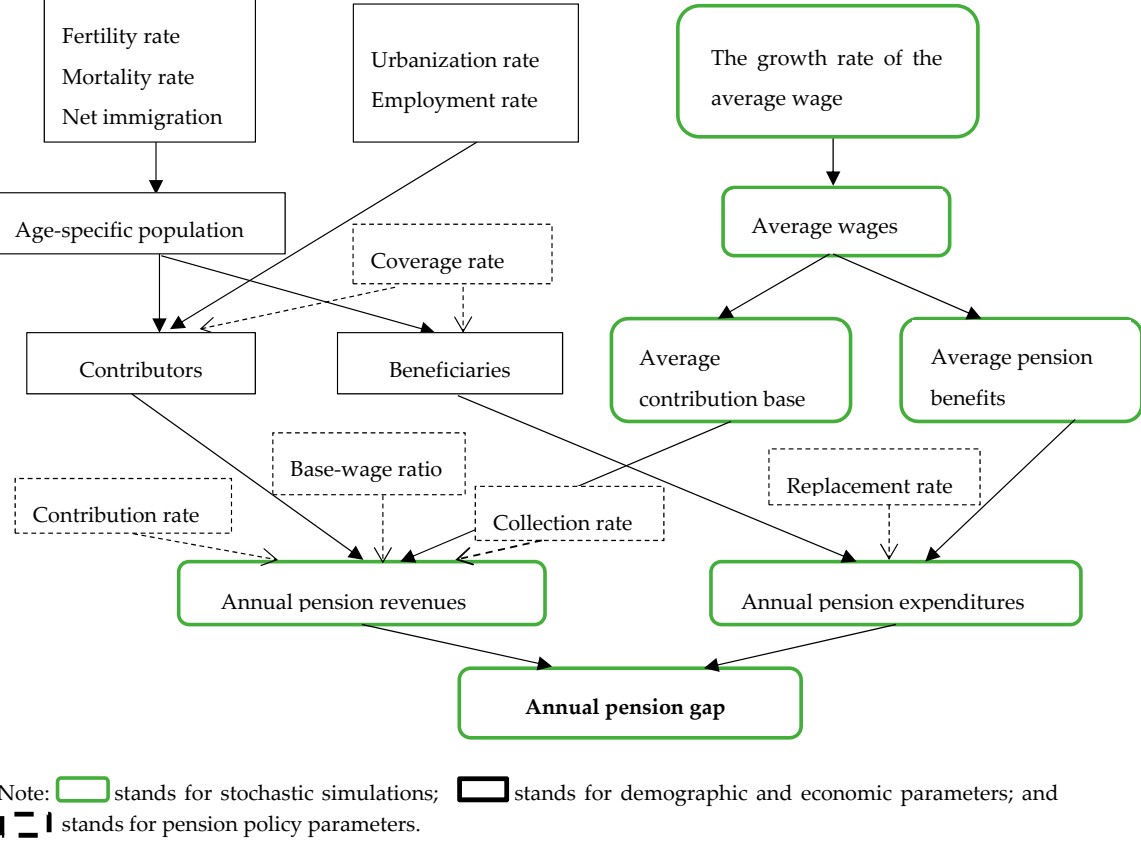

**Figure 1.** Influence path of variables on annual pension gap of the research.

## 2. The Major Public Pension System in China

### 2.1. Public Pension Policies in China

The current public pension system in China includes the Public Pension System for Urban Employees and the Public Pension System for Urban and Rural Residents. The Public Pension System

for Urban Employees, which was founded in 1997, has developed from the traditional retirement system under planned economy to social insurance system under market economy. At first it only covered the employees from urban enterprises and then expanded the coverage to non-standard employees in informal sectors from 2005. In 2015, the public pension system for civil servants has been merged into the urban employees' system. So far, public pension system covering all urban employees has been established. By the end of 2016, the number of participants in the system has reached 379 million. Meanwhile, the Public Pension System for Urban and Rural Residents, established in 2014, was an integration of the new pension system for rural residents founded in 2009 and the public pension system for urban residents in 2011. At present, the residents' pension system, covering 508 million participants, is mainly financed by fiscal subsidies and more like a welfare system, therefore it is not in the scope of our discussion. The Public Pension System for Urban Employees, featured as the social insurance, is the focus of this analysis.

With more than 20 years of reform, the Public Pension System for Urban Employees has been established with a combination of social pooling and individual account. According to the current policy, the social pooling part is a pay-as-you-go system financed by 20% of contribution wages from employers or 12% of contribution bases from self-employees, while individual account is financed by 8% of contribution wages from employees. However, in reality the individual account is not fully funded and usually mixed with social pooling part [27]. Therefore, we decide to calculate the pension gap by combining the two parts together.

## 2.2. Current Situation of the Public Pension System for Urban Employees

### 2.2.1. Participation Status

In order to ensure old-age income security for more labor force, the public pension system for urban employees has been continuously promoting universal coverage. The system has expanded its coverage to non-state-owned enterprise employees, self-employees, migration workers and other non-standard employees in informal sectors. By 2016, the total number of insured people has reached 379 million, which was three times that of the total number of the insured when the system was established in 1998. Among them, the number of insured employees rose from 84.75 million to 278 million and the coverage rate for insured employees gradually increased from 39.21% to 67.17% [28]. Although the government has made achievements in expanding coverage, some small and medium-sized enterprises and groups with low contribution capacity are not covered by the system.

As for the age structure of participants in the pension system, during the period of 1998 to 2016, the number of retirees increased from 27.27 million to 101 million [28] and the support ratio decreased from 3.11 to 2.75. That is to say, there are only 2.75 employees supporting one retiree nowadays, which reflects a serious aging problem within the pension system.

Concerning the composition of insured population, the proportion of non-standard employees as individual status participating in the pension system has increased from 18% in 2007 to 25% in 2015. With the integration of urban public pension systems, the number of enterprise employees has reached 254.92 million, accounting for 67.2% of the total number, while the number of non-standard employees has reached 87.72 million, accounting for 23.1% and civil servants has reached 36.66 million, accounting for 9.7% [29]. It is noteworthy that the statutory contribution rate of non-standard employees is lower than that of the participants from formal sectors. Figure 2 shows the composition of participants of urban public pension systems in recent years.

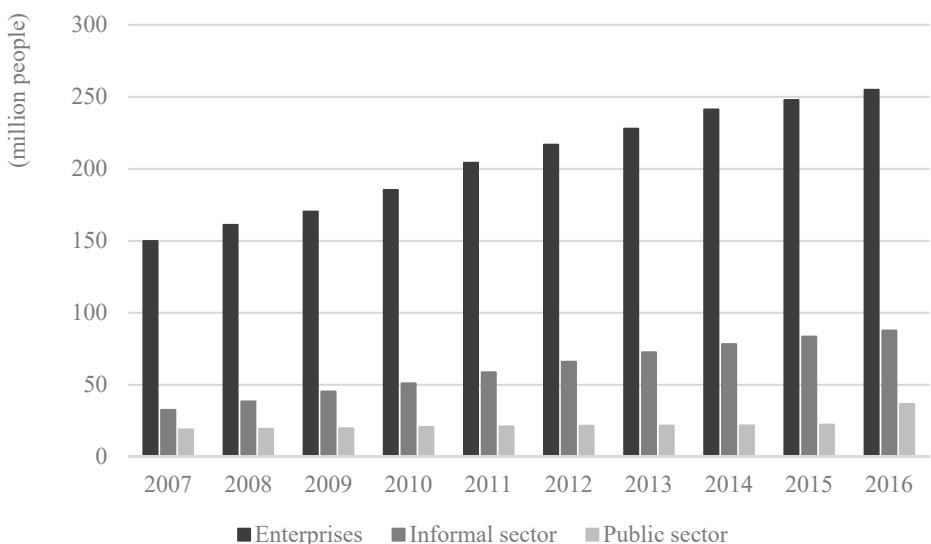

**Figure 2.** The type of insured population in the urban public pension system from 2007 to 2016.

### 2.2.2. Actual Situation of Pension Collection

According to the policy, the minimum contribution years for urban employees are 15 years and the contribution bases vary from 60% to 300% of the average wages. In practice, however, there are many cases of contribution in suspense, contribution with the lowest base and hence the actual contribution rate is far below the policy rate.

The first problem in pension collection is that the actual collection rate is far below 100%. In 2006, the collection rate, the ratio of actual contributors to insured employees, was around 90% but it continued to decline to less than 80% in 2016 [30]. That is to say, every one out of five insured employees fails to contribute. There are many reasons why the contribution is suspended. For instances, the pressure of economic downturn is increasing, the performance of enterprises is very low and many non-standard employees stop contributing as their contribution years reach the 15-year limit [31].

The second problem is that the contribution base is much lower than the average wage. As is stipulated in the 2011 Social Insurance Law, employers should contribute in proportion to the total wages of their employees and employees are required to contribute in proportion to their own wages. However, due to lack of supervision, many enterprises determine their contribution bases according to their own standards that benefit themselves most. According to the annual contribution record, the average contribution base accounts for 81.5% of the urban average wages in 2000 and the number falls to 72.2% in 2006 and further declines to 63% in 2015.

Under the influence of the former two factors, the third problem will inevitably come up: the actual contribution rate is much lower than the statutory contribution rate. A low contribution base accompanied with a low collection rate will surely result in a decline in contribution revenues. The actual contribution rate can be calculated by the ratio of average contributions to the average wages in urban sectors. According to statistics from 2006 to 2016, the actual contribution rate of public pension system for urban employees gradually declines from 19.06% to 15.86%, far lower than the statutory contribution rate in formal sectors (28%) and informal sectors (20%). Based on the survey conducted by the Chinese Academy of Fiscal Sciences, the proportion of pension contributions to total wages in 2015 is about 15.59% and the contribution rate varies greatly among regions: the average contribution rate in the eastern, central and western regions is 11.09%, 15.74% and 17.02% and the Northeast Region is as high as 22.07% [32].

Therefore, the projection about the balance of the future pension fund should be based on the practice of policy implementation rather than the policy regulations.

### 2.2.3. Pension Balances and Expenditure

In practice, statistical data reflecting the financial status of public pensions do not distinguish between social pooling and individual account but take them as a whole, which also reflects the mixed using of the pension system. According to statistical bulletins on the development of human resources and social security since 2000, Table 1 displays the balances of pension fund from 2001 to 2016. According to the table, the accumulated balance of pension fund has increased year by year, from 105.4 billion yuan at the end of 2001 to 3858 billion yuan in 2016. Meanwhile, the payable months of the accumulated fund increased from 5.45 months to 18.46 months in 2012, followed by a downward trend year by year, reaching 14.53 months in 2016 [33]. It is worth noting that apart from collection income and its interest within pension revenues, there are also a large number of financial subsidies from governments at all levels each year. When financial subsidies are excluded, there exists a pension gap in 2002. With the coverage expansion implemented in 2005, this gap reduced. However, the pension gap excluding financial subsidies comes up again in 2015 and 2016. Additionally, the annual growth rate of pension expenditures has exceeded the growth rate of pension revenues since 2012. Obviously, the financial balance of the pension fund in the future is not optimistic.

**Table 1.** The balance of fund in Public Pension System for Urban Employees (2001–2016).

| Year | Total Balances (100 Million Yuan) | Growth Rate of Total Balances (%) | Contributions (100 Million Yuan) | Interest Balances (100 Million Yuan) | Fiscal Subsidies (100 Million Yuan) | Total Expenditure (100 Million Yuan) | Growth Rate of Total Expenditure (%) | Accumulative Balance (100 Million Yuan) |
|------|------|------|------|------|------|------|------|------|
| 2001 | 2489 | 9.24 | - | - | - | 2321 | 9.73 | 1054 |
| 2002 | 3172 | 27.42 | 2551 | 165 | 455 | 2843 | 22.47 | 1608 |
| 2003 | 3680 | 16.03 | 3044 | 106 | 530 | 3122 | 9.82 | 2207 |
| 2004 | 4258 | 15.72 | 3585 | 59 | 614 | 3502 | 12.17 | 2975 |
| 2005 | 5093 | 19.61 | 4312 | 130 | 651 | 4040 | 15.37 | 4041 |
| 2006 | 6310 | 23.88 | 5215 | 124 | 971 | 4897 | 21.20 | 5489 |
| 2007 | 7834 | 24.16 | 6494 | 183 | 1157 | 5965 | 21.81 | 7391 |
| 2008 | 9740 | 24.33 | 8016 | 287 | 1437 | 7390 | 23.88 | 9931 |
| 2009 | 11,491 | 17.97 | 9534 | 311 | 1646 | 8894 | 20.36 | 12,526 |
| 2010 | 13,420 | 16.79 | 11,110 | 356 | 1954 | 10,555 | 18.67 | 15,365 |
| 2011 | 16,895 | 25.89 | 13,956 | 667 | 2272 | 12,765 | 20.94 | 19,497 |
| 2012 | 20,001 | 18.38 | 16,467 | 886 | 2648 | 15,562 | 21.91 | 23,941 |
| 2013 | 22,680 | 13.39 | 18,634 | 1027 | 3019 | 18,470 | 18.69 | 28,269 |
| 2014 | 25,310 | 11.60 | 20,434 | 1328 | 3548 | 21,755 | 17.79 | 31,800 |
| 2015 | 29,341 | 15.93 | 23,016 | 1609 | 4716 | 25,813 | 18.65 | 35,345 |
| 2016 | 35,058 | 19.48 | 26,768 | 1779 | 6511 | 31,854 | 23.40 | 38,580 |

With regard to the level of pension benefits, the replacement rate is generally used to examine pension adequacy. In China, the replacement rate of pension to average wages is often used because it can reflect residents' consumption level in the changing socio-economic environment. Since the establishment of the system in 1998, the average pension replacement rate has dropped from about 70% to around 44% in 2016. This is because the annual growth rate of average wages is larger than that of the average pension benefit, even though the government raises the pension level each year. Comparing with the standard set by ILO, we could think of current pension benefits as acceptable in average.

## 3. Basic Assumptions and Data Sources

### 3.1. Basic Assumptions

To assess the pension gap caused by contributions and the payment of benefits, financial subsidies are not taken into consideration in this analysis. This research sets 2016 as the base year and makes actuarial assumptions on demographic, economic and institutional parameters for the next 80 years.

- The age structure of insured population: due to lack of data on the age structure of the insured population, it can be assumed that the age structure of the participants is the same as that of the urban employees and the age structure of the retirees is the same as that of the urban elderly in corresponding years.

- Pensionable age: the statutory retirement age is usually equivalent to the pensionable age in China. According to the current policy, it can be assumed that the retirement age is 60 year-old for males and 55 year-old for females and the age for economically active population is between 16 and 59.

- Mortality rate: There are many mortality forecasting models among which some well-known models, such as De Moivre model, Gompertz model, Makeham model, Weibull model, were established in the early stage and failed to consider the change of death patterns over time. Later on, the Lee-Carter model, which considers the change of mortality with age and time, has been widely applied. Combining with China's reality, Wang and Ren [34] have established a mortality forecasting model based on Lee-Carter model with limited data. Using this model, we select the data of 10% population sampling survey in 1986, 1% population sampling survey in 1995 and 2005 and the population census data in 2000 and 2010 to predict the mortality rate of males and females by 2100. In addition, we assume that the age limit for the population is 101 years old.

- Fertility rate: The fertility rate is determined by the total fertility rate and the age-specific fertility rate. Considering the urbanization and the improvement of women's educational level, the fertility pattern is bound to change greatly in China's society. Since the population forecast made by United Nations has considered factors mentioned above, we adopt the assumptions on total fertility rates and age-specific fertility rates from the medium projection of 2017 World Population Prospects [35]. In China, apart from natural factors, economic and social conditions and traditional concepts, family planning policy also plays an important role in determining fertility level. Therefore, it is not suitable to establish stochastic time series model for fertility related indicators.

- Sex ratio at birth: Although there are many predictions made about sex ratio at birth in China, there is no widely-acknowledged authoritative projection. Therefore, the projection of the sex ratio at birth will be directly quoted from the 2017 World Population Prospects [35].

- Net immigration rate: we assume that the net immigration rate would be zero in the future because we mainly focus on the sustainability of the domestic pension system and do not rely on overseas immigrants to alleviate financial gap.

- Urbanization rate: The projection method constructed by Liu Shijin [36] from the "medium and long-term growth" research group in the Development Research Center of the State Council is used here and the logistic model designed to forecast the future urbanization rate is as follows.

$$\mathrm{U(t)} = \frac{0.569}{1 + 14.22exp(-0.098t)} + 0.17 \tag{1}$$

where t stands for year t.

- Urban employment rate: According to the China Statistical Yearbook over years, the urban employment rate has been remained at 85% for the past 20 years, so it is reasonably assumed that the urban employment rate in the future still remains at the same level.

- Coverage rate: The coverage rate of urban employees in China has increased from 39.2% in 1998 to 67.17% in 2016. We assume the future coverage rate still remains at this level.

- Structure of insured contributors: The insured contributors are composed of formal sector employees and informal sector labors. The formal sector employees refer to those whose contributions come from their employers as well as them, whereas the informal sector labors refer to those whose contributions only come from them. In 2016, the proportion of informal sector labors accounted for about one quarter of the insured employees. We assume that this ratio remained unchanged in the future.
- Contribution rate: The statutory contribution rate for formal sector employees is 28% and 20% for informal sector labors. Considering the relevant scales of the two types of contributors, we calculate the weighted statutory contribution rate as 28% × 75% + 20% × 25% = 26%. It is also assumed that the rate will remain unchanged in the future.
- Collection rate: In 2016, the collection rate, namely the ratio of contributors to the insured employees, is about 80%. We assume that the collection rate will remain unchanged in the future.
- Contribution base: According to the statistics of the labor department, the ratio of the average contribution base to the average wage is about 63% in 2016. We assume that the contribution base as a percentage of the average wage will remain unchanged in the future.
- Pension replacement rate: In 2016, the average pension for urban employees is about 2362 yuan per month, accounting for 44.8% of the average monthly wage (5270 yuan) for urban employees in the previous year. It is assumed that the pension replacement rate in the future remains constant.

### *3.2. Data Sources*

- Initial population: The age-specific population of males and females in 2016 is estimated from the Sixth Population Census in 2010. Due to the under-reporting that may occur in younger age group, this paper adjusts the number of the younger age according to school entry indicators.
- Public pensions: The data of pension participation and fund balances is based on China Labor Statistics Yearbook, Statistical Bulletin on the Development of Human Resources and Social Security and Annual Report of the Social Insurance Development in China.
- Other indicators: Other macro data, such as the participation of labor force, average wages for urban employees, comes from the China Statistical Yearbook.

## 4. Projection for Population and Pension Gap

In this section, we first forecast age-specific population on the basis of demographic assumptions and then predict working age population and employees in urban areas. Next, contributors and pensioners can be estimated according to assumptions of coverage rates. Finally, models of annual pension balance could be constructed.

### *4.1. Projection for Population*

#### 4.1.1. Forecasting Model

By the use of Cohort-component population projections, this paper establishes a forecasting equation for population:

$$P(t + n) = P(t) + B(t) - D(t) + I(t) - E(t) \qquad (2)$$

where $P(t)$ is the population at the year $t$, $B(t)$ and $D(t)$ are number of births and deaths occurring between $t$ and $t + n$, $I(t)$ and $E(t)$ are the number of immigrants and of emigrants from the country during the period $t$ to $t + n$. Since we assume that the net immigrate rate equals to zero, the annual increase in population is mainly determined by the birth rate and mortality rate.

By adopting Leslie matrix, we can calculate the population of the next year based on that of the current year. The specific method is as follows:

$$
\begin{bmatrix} n_0 \\ n_1 \\ \vdots \\ n_{\omega-1} \end{bmatrix}_{t+1} = \begin{bmatrix} f_0 & f_1 & f_2 & \cdots & f_{\omega-2} & f_{\omega-1} \\ s_0 & 0 & 0 & \cdots & 0 & 0 \\ 0 & s_1 & 0 & \cdots & 0 & 0 \\ 0 & 0 & s_2 & \cdots & 0 & 0 \\ \vdots & \vdots & \vdots & \ddots & \vdots & \vdots \\ 0 & 0 & 0 & \cdots & s_{\omega-2} & 0 \end{bmatrix} \begin{bmatrix} n_0 \\ n_1 \\ \vdots \\ n_{\omega-1} \end{bmatrix}_t \tag{3}
$$

In this equation, matrix $\begin{bmatrix} n_0 \\ n_1 \\ \vdots \\ n_{\omega-1} \end{bmatrix}_{t+1}$, denoted by $n_{t+1}$, stands for population of all age groups in the year of t + 1; and $\begin{bmatrix} n_0 \\ n_1 \\ \vdots \\ n_{\omega-1} \end{bmatrix}_t$, denoted by $n_t$, represents population of all age groups in the year of t; and $\begin{bmatrix} f_0 & f_1 & f_2 & \cdots & f_{\omega-2} & f_{\omega-1} \\ s_0 & 0 & 0 & \cdots & 0 & 0 \\ 0 & s_1 & 0 & \cdots & 0 & 0 \\ 0 & 0 & s_2 & \cdots & 0 & 0 \\ \vdots & \vdots & \vdots & \ddots & \vdots & \vdots \\ 0 & 0 & 0 & \cdots & s_{\omega-2} & 0 \end{bmatrix}$ is the Leslie matrix, denoted by **L**. f indicates the fertility level of women. The fertility level for women aged between 15 and 49 is determined by both the total fertility rate and the age-specific fertility rate and the fertility rate of women in other age groups is 0. S indicates the survival probability and $\omega$ is the age limit. Therefore, the forecasting equation of population can be simplified as the following form:

$$
n_{t+1} = \mathbf{L} \times n_t \tag{4}
$$

Next, the number of gender-specific newborn babies in each year can be calculated based on the assumption of the sex ratio at birth. Then the number of gender-specific and age-specific populations can be estimated.

### 4.1.2. Forecast of Future Population

It is projected that China's total population will peak in 2030, reaching nearly 1.5 billion and decline gradually to 1.41 billion in 2050 and 1.04 billion in 2100. Moreover, the old-age dependency ratio increases gradually from 16% in 2017 to more than 50% in 2055, reaching the peak of population aging and remains at above 45% in the future. Figure 3 displays the projection for future contributors and pensioners. It can be seen that the total insured population increases from 386 million in 2017 to 510 million in around 2045 and then declines to about 370 million by the end of the century. Besides, it is projected that only 1.5 employees support one retiree by the middle of the century compared with the fact that about 3 employees support one retiree at the moment.

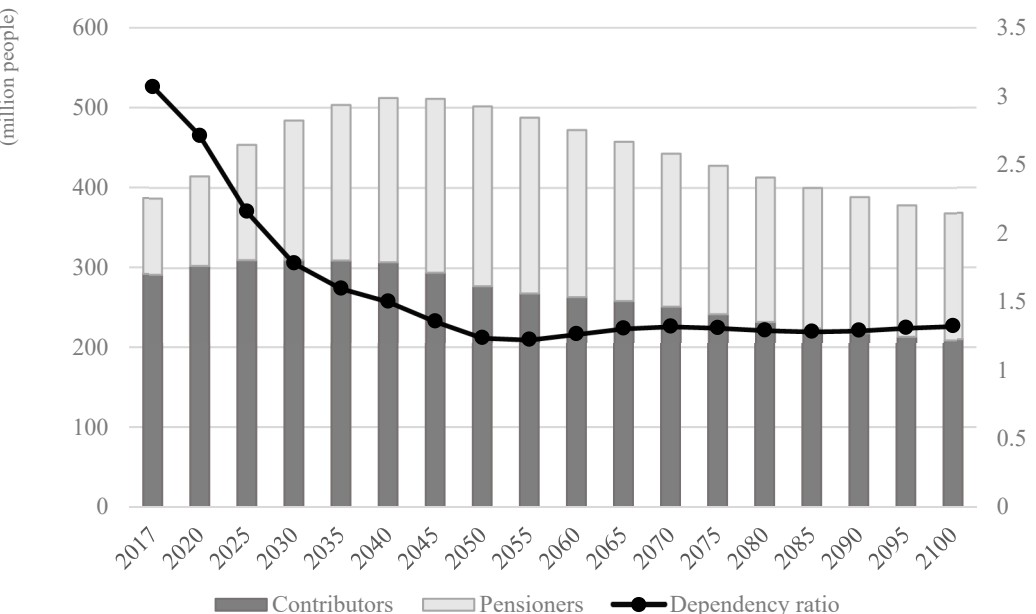

**Figure 3.** Forecast of insured population in China from 2017 to 2070.

### 4.2. Projection for Pension Gap

#### 4.2.1. Stochastic Time Series Model for Wage Growth Rate

Using the data of the growth rate of the average wage over the past 60 years, we are able to build a time series model to predict its future changes. After taking Augmented Dickey-Fuller test and comparing the estimated results of various AR models, AR(1) model is selected to fit the sequence. Let the growth rate of the average wage in year t be $g_t$, the error term is $\varepsilon_t$, the estimated time series model is:

$$g_t = 0.0191 + 0.576g_{t-1} + \varepsilon_t, \ \varepsilon_t \sim N\left(0, \ 0.0537^2\right) \tag{5}$$

The sequence of the residuals has passed the White-Noise test. Then the AR(1) model is appropriate to fit the time series. When extracting a series of random numbers from the normative distribution $N\left(0, \ 0.0537^2\right)$ as $\{\varepsilon_t\}$, the predicted values of wage growth rates are produced. After 5000 Monte Carlo simulations, the probability distribution of future wage growth rates is displayed in Figure 4. According to the simulation, we construct 95% confidence interval and find that by 2070 the values in the 5% percentile, the median and the 95% percentile are 3.85%, 4.32% and 4.78% respectively.

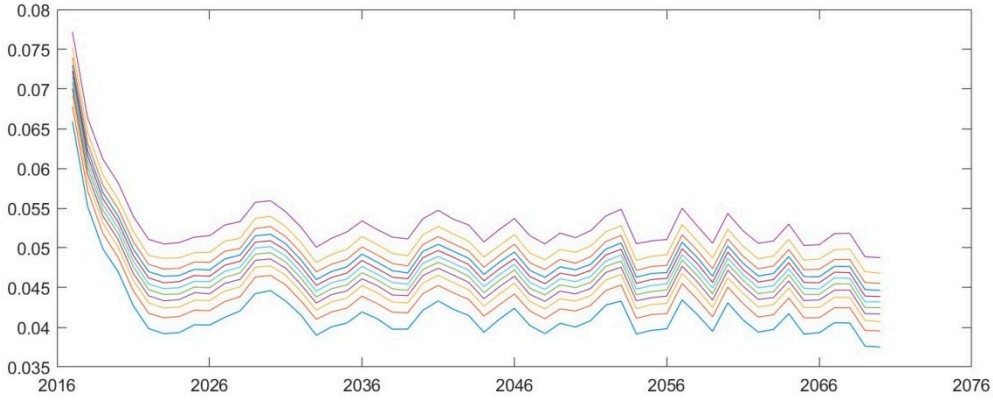

**Figure 4.** Simulation of future growth rates of average wages.

4.2.2. Forecasting Model for Pension Gap

The flow-based calculations for yearly pension gap are the difference between annual revenues and expenditures.

The model of pension revenues is in the following:

$$PC(t) = Emply(t) \times CovRate(t) \times ConBase(t) \times ConRate(t) \times CollRate(t) \tag{6}$$

$$ConBase(t) = g_t \times AveWage(t-1) \times \gamma(t) \tag{7}$$

where PC(t) represents the pension revenues collected in the year of t, Emply(t) represents the number of urban employees in year t, CovRate(t) is the coverage rate of the public pension system for urban employees in year t, ConBase(t) is the average contribution base in year t, ConRate(t) stands for the contribution rate in year t CollRate (t) stands for the collection rate in year t, AveWage(t − 1) stands for the average wage of urban employees in the year of t − 1, $\gamma(t)$ is the ratio of the average contribution base to the average wage in year t.

The model of pension expenditure is as follows:

$$PE(t) = Retiree(t) \times ReplRate(t) \times AveWage(t-1) \tag{8}$$

$$AveWage(t) = AveWage(2016) \times \prod_{t=2017}^{2070} g_t \tag{9}$$

where PE(t) represents the pension expenditure in year t, Retiree(t) represents the number of pensioners in year t, ReplRate(t) represents the average replacement rate in year t.

Therefore, the model of the yearly pension gap is as below:

$$PG(t) = PC(t) - PE(t) \tag{10}$$

where PG(t) stands for pension gap in year t. When PG(t) < 0, it indicates that there is a pension gap in year t and the amount of the pension gap equals to −PG(t); when PG(t) ≥ 0, it indicates that there is no gap in year t.

4.2.3. Forecast of Future Pension Gap

Figures 5–7 depict the probability distribution of annual pension revenues, pension expenditures and pension gap within the 95% confidence intervals from 2017 to 2070 under basic assumptions. As shown by the figures, the fluctuation degree of all the three variables increases over time but the fluctuation degree of pension expenditures changes faster than that of pension gap and pension revenues. The 95% confidence intervals of the three variables are basically symmetrical with respect to the means. By the year of 2070, the average pension revenue is projected to be 1.80 trillion yuan with a standard deviation of 0.11 trillion yuan and the average pension expenditure is 5.16 trillion yuan with a standard deviation of 0.31 trillion yuan, while the average pension gap is 3.36 trillion yuan with a standard deviation of 0.2 trillion yuan. It can be seen that in a certain period of time, the fluctuation degree of pension expenditures and pension gap is higher than that of pension revenues.

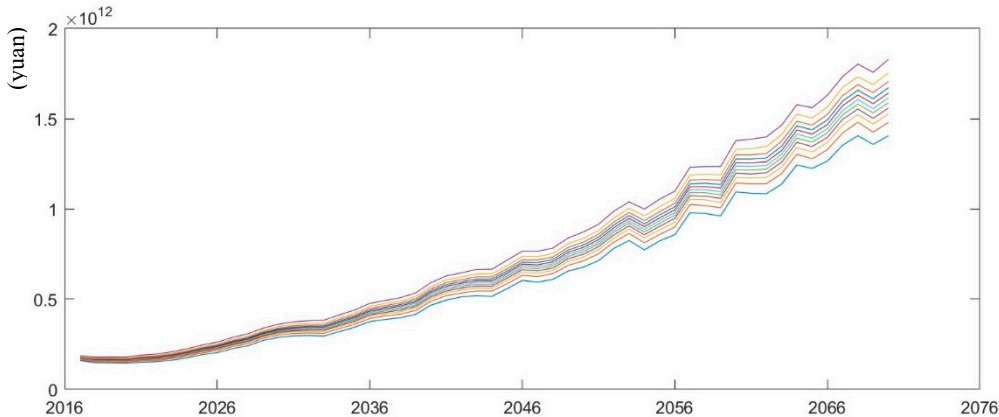

**Figure 5.** Distribution of future pension revenues under basic assumptions.

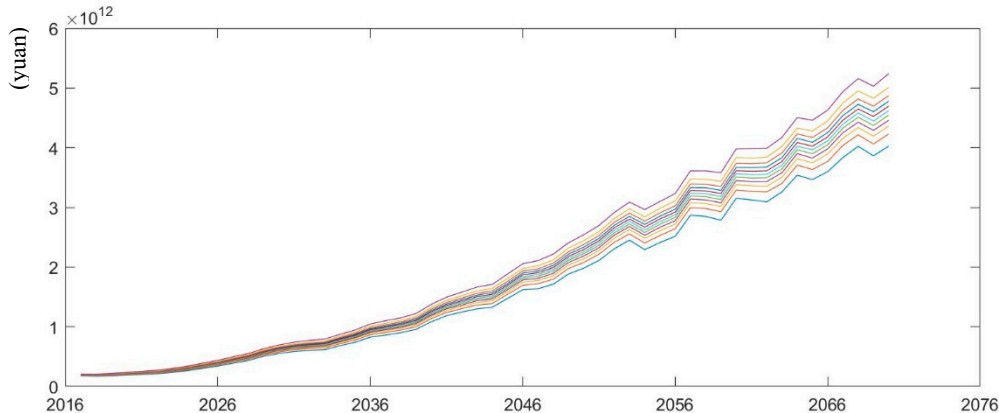

**Figure 6.** Distribution of future pension expenditures under basic assumptions.

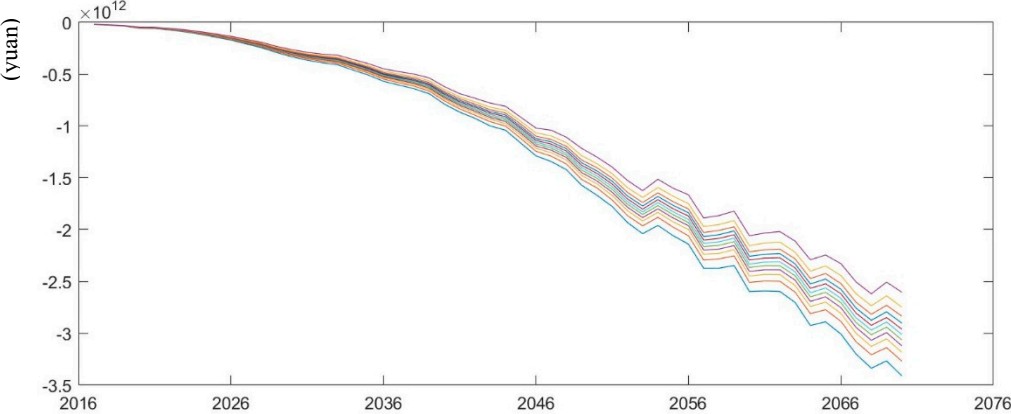

**Figure 7.** Distribution of future pension gap under basic assumptions.

## 5. Sensitivity Analysis

The above calculations are based on the baseline assumptions. If the actuarial assumptions change, the pension gap will produce different results. To analyze the impact of changes in actuarial assumptions on the projection results, we use sensitivity analysis. Many studies have examined the impact of changes in demographic and macroeconomic factors on the change of projections [17] and [37] and some have concerned about the impact of postponing the retirement age on the changing results [11,20]. However, less attention has been paid to the impact of policy parameters. This paper will examine the impact of mortality rates as well as combination of policy parameters.

### 5.1. Scenario Analysis on the Mortality Rate

According to the mortality forecasting models in various scenarios of Wang and Ren (2012) by [34], we simulate pension gap under low and high mortality rates from 2017 to 2070. As is shown in Figure 8; Figure 9, future pension gap becomes larger under low mortality rate compared with the scenario of high mortality rate. When the median values of pension gap under various scenarios are extracted, the changes in pension gap over time are easily observed. According to Table 2, in 2050 and 2070, the pension gap under the low mortality rate is projected to increase by 9.74% and 0.79% compared with the baseline scenario, whereas the pension gap under the high mortality rate will decrease by 8.76% and 12.9% respectively compared with the medium mortality rate. This phenomenon can be explained by the fact that longevity prevails, and the number of pensioners increases under the low mortality rate, leading to an increase in pension expenditures without increasing any contributions. Thus, the sustainability of pension fund deteriorates. The situation becomes the opposite under the high mortality rate.

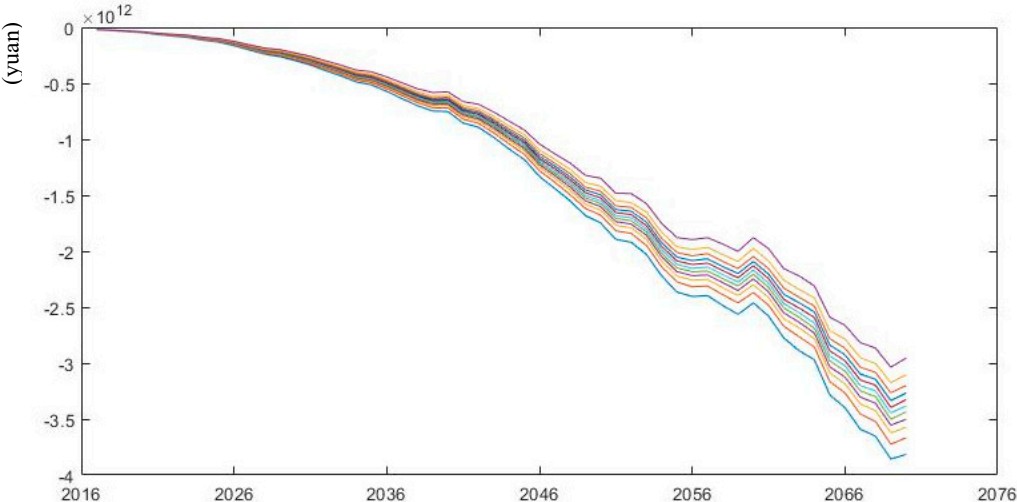

**Figure 8.** Pension gap under the low mortality rate.

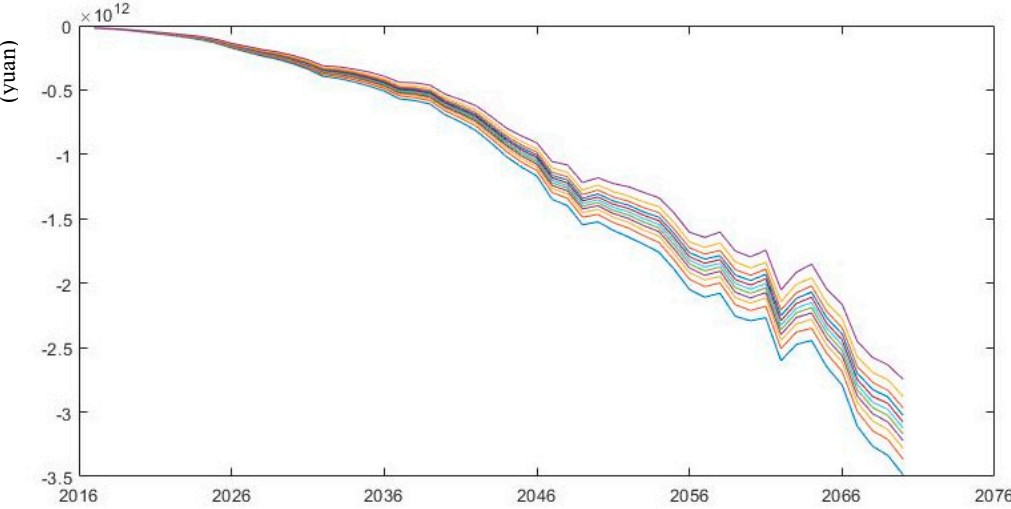

**Figure 9.** Pension gap under the high mortality rate.

**Table 2.** Median pension gap under various mortality rates.

| Assumptions on Mortality Rate | Median Pension Gap (Trillion Yuan) | | |
| --- | --- | --- | --- |
| | 2020 | 2050 | 2070 |
| Baseline (medium) | −0.053 | −1.413 | −3.355 |
| Low | −0.044 (−17.25%) | −1.551 (9.74%) | −3.382 (0.79%) |
| High | −0.044 (−16.85%) | −1.290 (−8.76%) | −2.922 (−12.90%) |

Note: Numbers in brackets indicate the change in the ratio of pension gap under extended scenarios compared with the baseline scenario.

## 5.2. Scenario Analysis on the Statutory Retirement Age

Extending the legal retirement age is one of the policy measures government would like to consider for the purpose of pension sustainability. However, the policy is always challenged with political barriers, such as strikes and social anxiety. This is also the case for China. In recent years, the idea of gradually delaying retirement age has entered the government agenda but soon the proposal collapsed because of enormous pressure from the public and social media.

Taking into account the increasing trend of life expectancy and longevity risk in China, we simulate the change of pension gap when the legal retirement age is gradually increased by 1–2 years in the future. According to the results in Table 3, holding other parameters constant, if the legal retirement age of males and females are extended by 1 year in 2030, which means the retirement age for men is 61 and for women is 56 and the legal retirement age is further extended by 2 years in 2040, which means 62 for men and 57 for women, the median pension gap will be 15.95%, 8.04% and 17.80% smaller than that of the baseline scenario in 2020, 2050 and 2070 correspondingly. As for the cause, the increase of pensionable age means the increase in the number of contributors and the decrease in the number of pensioners at the same time point, which leads to the increase of pension revenues and the decrease of pension expenditures, thus reducing the pension gap.

**Table 3.** Median pension gap under extended statutory retirement age.

| Assumptions on Statutory Retirement Age | Median Pension Gap (Trillion Yuan) | | |
| --- | --- | --- | --- |
| | 2020 | 2050 | 2070 |
| Baseline (60/55) | −0.053 | −1.413 | −3.355 |
| Extend 1–2 year from 2030 | −0.045 (−15.95%) | −1.300 (−8.04%) | −2.758 (−17.80%) |

Note: Numbers in brackets indicate the change in the ratio of pension gap under extended scenarios compared with the baseline scenario.

## 5.3. Scenario Analysis on Combination of Policy Parameters

In the context of delayed retirement age, the following part examines the net impact of different policy combination on future pension gap.

Firstly, in terms of the coverage, the universal coverage is one of the basic goals in establishing the public pension system. Since 2005, the Public Pension System for Urban Employees has not only covered the enterprise employees but also extended its coverage to self-employees and non-standard labor force. The 13th Five-Year Plan on Social Security has promoted universal participation plan, which encourages participation of new-type employment in electronic era. According to statistics, most of the urban employees have participated into the pension system, whereas many non-standard labor forces in informal sectors are not covered by the system [38]. If the system continues to expand its coverage, the proportion of the insured participants as non-standard labor force in the future will probably increase. According to the international standard, coverage rate of 90% is deemed as

universal coverage. Taking China's situation into consideration, we simulate changes in the pension gap when the coverage rate gradually reaches 80%, 85% and 90% in 2030 and then keeps unchanged. The coverage between 2017 and 2030 is obtained by linear interpolation. Along with the coverage expansion, the proportion of non-standard employees in all contributors is assumed to increase from 25% (baseline) to 30%, 35% and 40% accordingly.

Secondly, pension collection is of hot debate nowadays. According to *The Reform Plan on Collection and Management System of the National and Local Taxations* issued in July 2018, all contributions of social insurances will be collected by tax authorities from January 2019 [39]. Feng (2013) has argued that contributions collected by tax authorities instead of social security agencies can substantially increase the contribution revenues especially from non-state-owned enterprises [40]. With regard to the collection rate, it reflects not only the degree of standardization in pension collection but also the degree of continuity in contributions. With the reform of collection agencies of social insurances, it will not be allowed to evade contributions or contribute by a lump-sum. Therefore, it is reasonably assumed that the collection rate will gradually increase from 80% (baseline) to 85%, 90% or 95% in 2030. Considering the contribution base, it is a key factor in determining the pension revenues. With the standardization of premium collection, the proportion of contribution bases in average wages will increase accordingly. It is assumed that the ratio of the contribution base to the average wage starts to increase to 80%, 90% or 100% from 2030 and keeps stable since then. The ratio between 2017 and 2030 can be calculated by linear interpolation.

However, under the pressure of economic downturn in China, the reform on pension collection has aroused strong concerns from enterprises about labor costs and their competitiveness. Later on, the State Council emphasizes that the primary task of the current work is to make stable collection. Before the tax collection authorities are in place, it is strictly forbidden for all authorities to collect arrears [41]. The government is considering lowering contribution rates in the next step of reform. In 2016, the policy of reducing contribution rates of social insurances at stages was promulgated by the Ministry of Human Resources and Social Security and the Ministry of Finance. In April 2018, the two ministries issued another regulation on further reducing contribution rates of social insurances. At present, several provinces have set the contribution rate for employers as 19%. The feasible contribution rate with the range of 15% to 18% has been estimated based on affordability of individuals and enterprises [42–44]. According to this, the paper assumes that the average contribution rate of the public pension system will be gradually reduced from current 28% for standard employees and 20% for non-standard employees to 19%, 17% or 15% for both in 2030.

Last but not the least, this paper assumes the current pension benefit, 44% replacement of average wages as a basic level of protection and future policy reform should be based on the premise of not reducing the current protection level. This is roughly set by the Social Security (Minimum Standards) of ILO Convention No. 102, saying that a typical adult-male workforce with a spouse who has contributed for 30 years should have a replacement rate of no less than 40% [45]. Nevertheless, further research on the appropriate level of protection is needed in the future.

Usually, policy parameters are correlated with each other. Generally speaking, the lower the threshold of the pension system, the wider the coverage will be [46]. Plan A to Plan C is a combination of policy parameters under different coverage rates. Under the largest coverage in Plan C, the collection rate, the contribution rate and the ratio of contribution base to average wage are the lowest; while under the smallest coverage in Plan A, the other policy parameters are the highest; and the Plan B is in the between. Figures 10–12 show the distribution of pension gaps within 95% confidence interval under different combination of policy parameters. Pension gap under Plan A demonstrates the smallest value, followed by Plan B and Plan C. Further comparison of median pension gap in Table 4 reveals that compared with the baseline scenario, the median pension gap in Plan A is decreased by 61.26%, 18.32% and 23.05% in 2020, 2050 and 2070; and the median pension gap in Plan B is reduced by 27.83% and 7.93% in 2020 and 2070; however, Plan C increases median pension gap over the years. That is to say the wider the coverage, the worse the financial situation of the pension system. This could be

explained by the fact that in China the coverage expansion is conditioned on lowering policy threshold for low income group, so the long-term sustainability will be challenged if the formula for calculating pension benefits remains unchanged.

**Table 4.** Median pension gap under various combinations of policy parameters.

| Policy Combination | Baseline | Plan A | Plan B | Plan C |
|---|---|---|---|---|
| Coverage rate | 67% | 80% | 85% | 90% |
| Contribution base-average wage ratio | 63% | 100% | 90% | 80% |
| Collection rate | 80% | 95% | 90% | 85% |
| Contribution rate (weighted) | 26% | 19% | 17% | 15% |
| Statutory retirement age | 60/55 | Extend 1–2 year from 2030 | | |
| **Median Pension gap (trillion yuan)** | | | | |
| 2020 | −0.053 | −0.021 (−61.26%) | −0.038 (−27.83%) | −0.054 (1.27%) |
| 2050 | −1.413 | −1.155 (−18.32%) | −1.533 (8.44%) | −1.898 (34.27%) |
| 2070 | −3.355 | −2.581 (−23.05%) | −3.089 (−7.93%) | −4.093 (22.01%) |

Note: Numbers in brackets indicate the change in the ratio of pension gap under extended scenarios compared with the baseline scenario.

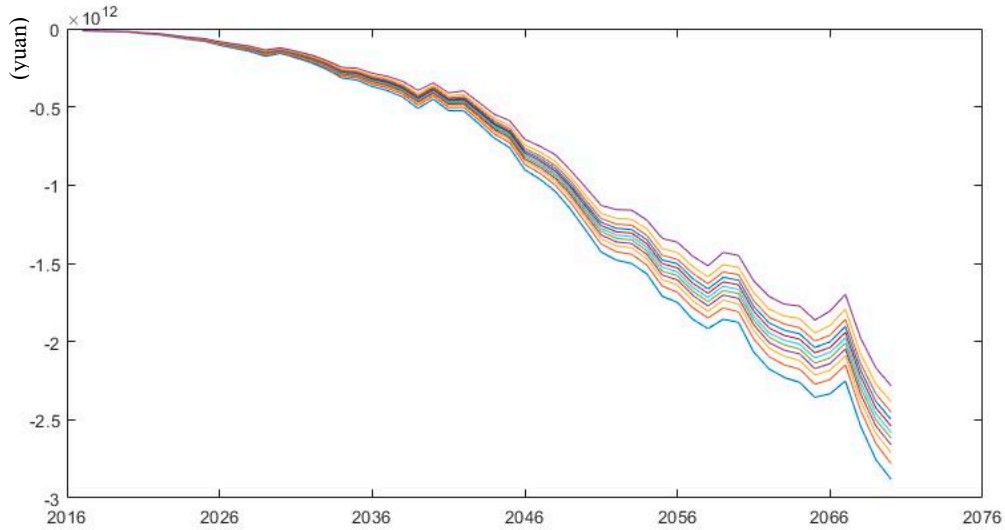

**Figure 10.** Pension gap under Plan A.

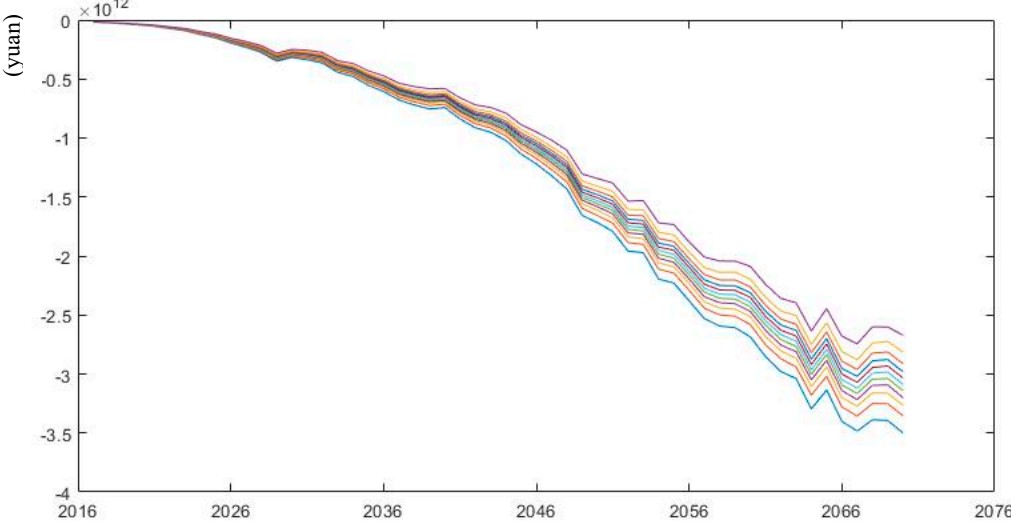

**Figure 11.** Pension gap under Plan B.

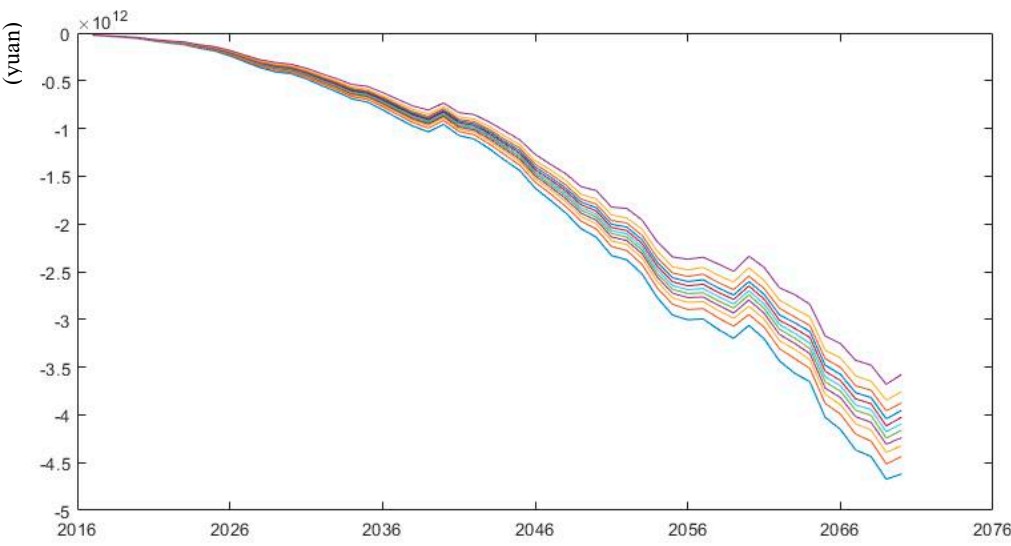

**Figure 12.** Pension gap under Plan C.

## 6. Conclusions

Under the circumstances of population aging and economic downturn, the sustainability of pension system has become a major concern for governments all around the world. This paper estimates the annual pension gap of the public pension system for urban employees in China. The projection of the future pension gap is more based on the current situation of the pension system, especially the problems brought by dual employment structure, differential contribution rates and insufficient collection. The stochastic prediction results show that, based on basic actuarial assumptions, if the current policy parameters remain unchanged, the pension gap will continue to exist and expand gradually from 2017. The median annual pension gap will reach 1.41 trillion yuan in 2050 and 3.36 trillion yuan in 2070. Sensitivity analysis of mortality rates suggests that the pension gap under the low mortality rate is projected to increase while decrease under the high mortality rate compared with the baseline scenario. Next, we make sensitivity analysis of combined policy parameters under the background of reform trend in China. The changes in policy parameters are generally interrelated and coverage expansion is usually accompanied by lower policy thresholds. Simulations of different reform plans show that pension sustainability will be improved with limited expansion, while the financial situation of the system under universal coverage is challenged with long-term unsustainability. As a consequence, the pension system for urban employees is facing two reform choices. First, for low-income group including a large number of non-standard employees with insufficient affordability, the government needs to adopt other systems, such as encouraging them to participate in the more redistributive basic old-age pension system for urban and rural residents or provide social assistances as safety nets for them. In short, universal coverage is achieved by multi-pillar systems. Second, more and more government financial subsidies will be invested in the unified pension system for the sake of basic protection.

**Author Contributions:** Q.Z. designed the project, established the model and wrote the original draft; H.M. contributed analysis tools and discussion and modified the final manuscript.

**Funding:** This research was funded by Youth Fund Project of Humanities and Social Sciences of the Ministry of Education (18YJC630259) and China Postdoctoral Science Foundation (2018M642922).

**Acknowledgments:** We would like to thank Zhen Li from School of Public Administration and Xiaojun Wang from School of Statistics, Renmin University of China for their insightful ideas on sustainable development of China's public pension system, which have given great inspirations to this article.

**Conflicts of Interest:** The authors declare no conflict of interest.

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
