# Peer review of "Evaluation on the Sustainability of Urban Public Pension System in China"

_sustainability, doi:10.3390/su11051418_

Round 1
Reviewer 1 Report
Referee report about the paper "Evaluation on the Sustainability of Urban Public Pension System in China", submitted by Qing Zhao and Haijie Mi for publication in Sustainability.
This paper deals with the sustainability of the public pension system for urban employees in China. The authors consider the indicator ‘annual pension gap’ as a good measure of the pension system sustainability. The paper analyzes the current situation of the pension system and introduces terminology and variables in order to project the population and forecast the pension gap. A sensitivity analysis under various scenarios of the pension system shows the effects on the pension gap.
The topic of the paper about the sustainability of a pension system is very interesting nowadays. A model to forecast the pension gap is considered, but the contribution with respect other models in the literature is not explained. For instance, [17] and [18] analyze the sustainability of a pension system but the relation with the current paper must be explained. Only a sentence in the introduction is included. There exist other papers with similar models? On the other hand, the methodology used only seems suitable for one of the public pension systems in China. Can be extrapolated to pension systems of other countries?
Although I think that the topic of the paper is interesting for the journal, in my opinion, the paper does not deserve publication in the journal in the current presentation.
Comments:
- The abstract is very long. The conclusions should not be included in it.
- In the introduction is anticipated that “This paper aims to establish actuarial models on the basis of the actual situation of pension policy, to make a more realistic forecast of annual pension gap, and to further explore the institutional factors affecting the sustainability of public pension system in China.”, but along the paper only a forecasting model is considered without cite previous models in the literature and the contribution with respect to them.
- The results of the Section 2 seem to be taken from official statistics, [23], [24] and [28], and then, it must not be considered as a contribution.
- The paper includes a forecasting model but without previous references to the specialized literature. If the forecasting model is not completely new, it should be indicated. The dynamic model does not include any prediction or probabilistic error, neither confidence interval, as in time series. The authors assume that the goodness of their model at 2017 and then it is used until 2098, without taking into account a prediction error or possible changes along the next years. Thus the model is some weak and I think that 80-years prediction is too much. The validity of the model would be explained in the paper.
- The sensitivity analysis is interesting. However, the time horizon is too long.
- The conclusions section includes sentences very similar to those in the abstract, and one literally taken from the Section 4.2 in relation to prediction results.
- The paper would substantially improve adding a new section where the authors suggest changes in the demographic and economic policies and analyze, with their model, the implications on the pension gap evolution, waiting that it improves. See, for instance, Section 4 in Li et al. (2015).
References
Liu, X., Zhang, Y., Fang, L., Li, Y. and Pan, W. (2015). Reforming China’s Pension Scheme for Urban Workers: Liquidity Gap and Policies’ Effects Forecasting, Sustainability 7(8), 10876-10894.
Author Response
Comments & Responses
1. The abstract is very long. The conclusions should not be included in it.
Thank you for your suggestion.We have revised the abstract.
2. In the introduction is anticipated that “This paper aims to establish actuarial models on the basis of the actual situation of pension policy, to make a more realistic forecast of annual pension gap, and to further explore the institutional factors affecting the sustainability of public pension system in China.”, but along the paper only a forecasting model is considered without cite previous models in the literature and the contribution with respect to them.
Many thanks for this important comment. In the revised version, we have added literatures of the model, OCA Stochastic Model (OSM for short) introduced by US Social Security Administration, on the basis of which we build the flow-based pension gap model. The OSM is a stochastic actuarial prediction model. It estimates the probability distribution of the future financial variables of the fund by considering the random fluctuation of one or more input variables (SSA, 2004; CBO, 2005). The OSM usually makes 75-year prediction for the US social security system and it is widely used around the world. Buffin,K.G.(2007) summarizes the basic ideas of stochastic actuarial prediction model as follows: based on the historical data of each input variable, the time series model is constructed first, then the Monte Carlo simulation method is used for random simulation, and finally the probability distribution of pension payment gap is obtained. For more details, please kindly check the revised paper in line 66-91. Finally, a figure of Influence path of the model design is added in section 1.
Concerning our model, we follow the basic method by constructing pension revenue and expenditure model based on parameters of demographics, economics and pension system. Compared with previous research, we improve the model by distinguishing standard employees (with the regulated contribution rates of 28%) and non-standard employees (with the regulated contribution rates of 20%). Further, we calculate the weighted regulated contribution rate according to the scale of non-standard employees. In addition, with regard to the collection rate, it not only reflects the standardization of pension collection, but also reflects continuity of contributions. In China, many people stops contributing as long as they reach the minimum contributing years of 15, which is extremely harmful to the sustainability of public pension system. However, all these important policy parameters are neglected in previous research studying Chinese pension sustainability ([22-25]). Thus the modified model aims to be fit for the reality of Chinese society.
3. The results of the Section 2 seem to be taken from official statistics, [23], [24] and [28], and then, it must not be considered as a contribution.
Table 1 is a combined statistics reflecting yearly balance of the public pension system for urban employees. Though it is not original contribution, as a kind of descriptive statistics, it presents the historical trend for readers to form an overall impression about pension financial status in China.
4. The paper includes a forecasting model but without previous references to the specialized literature. If the forecasting model is not completely new, it should be indicated. The dynamic model does not include any prediction or probabilistic error, neither confidence interval, as in time series. The authors assume that the goodness of their model at 2017 and then it is used until 2098, without taking into account a prediction error or possible changes along the next years. Thus the model is some weak and I think that 80-years prediction is too much. The validity of the model would be explained in the paper.
This is a really great point. In this revised version, we have constructed stochastic prediction model by introducing time series model of wage growth rate into the simulation, including probabilistic error and confidence interval to make interval estimation instead of point estimation, so as to overcome the non-robustness of prediction. Also, we have decreased the projection to 50 years.
5. The sensitivity analysis is interesting. However, the time horizon is too long.
Yes, the revised version have cut down the projection year a bit.
6. The conclusions section includes sentences very similar to those in the abstract, and one literally taken from the Section 4.2 in relation to prediction results.
Thank you for your suggestion. We have modified the abstract.
7. The paper would substantially improve adding a new section where the authors suggest changes in the demographic and economic policies and analyze, with their model, the implications on the pension gap evolution, waiting that it improves. See, for instance, Section 4 in Li et al. (2015).
This is a very important suggestion. We have added demographic factors in the sensitivity parts and economic factor in the stochastic modeling. We have included Li et al.(2015).

Reviewer 2 Report
The article assesses mainly fiscal sustainability of the Urban Pension System in China. It would be good to indicate that sustainability can be also measured using social approach, that is whether the system protects its beneficiaries from poverty and provides adequate income replacement. While it is not a subject of the article, it would be good to clearly mention this part.
I would also like to suggest to further discuss the scenario of expanded coverage. While it seems to affect the fiscal sustainability, it actually provides more social sustainability, as it increases the social protection of urban workers.
The simulations that are presented clearly present potential results of different policy options. In the comparison (part 5.4), as I understand, the results of all scenarios (but as separate reforms) are presented. It would be interesting to see what happens if all proposed measures are introduced in Urban Pension System, as some of the measures may interact, increasing or decreasing the total impact of the changes.
I have also found some editorial mistakes:
line 33, there should be plural form of the verb (point out)
line 93, there should be singular form for the verb (has)
line 120, the pension system does not have an age structure, it is rather the participants, in this particular context it is the comparison of contributors and beneficiaries.
Author Response
Comments & Responses:
1. The article assesses mainly fiscal sustainability of the Urban Pension System in China. It would be good to indicate that sustainability can be also measured using social approach that is whether the system protects its beneficiaries from poverty and provides adequate income replacement. While it is not a subject of the article, it would be good to clearly mention this part.
This is a very good point. In this version I have added in Section 5 (line 573-578) that current pension replacement rate of the average wage (44%) could be seen as a minimum level according to relevant standard set by ILO, meanwhile responding to the development of sustainability from financial sustainability to social sustainability (line 42-47) in the introduction part. Moreover, in the baseline scenario and sensitivity analysis, the research assumes that the replacement rate keeps stable and the contribution rate falls into the range between 15%-19%, thus to keep balance among financial sustainability, pension adequacy and affordability.
2. I would also like to suggest to further discuss the scenario of expanded coverage. While it seems to affect the fiscal sustainability, it actually provides more social sustainability, as it increases the social protection of urban workers.
This is also a good suggestion. The revised version have added three scenarios of coverage expansion. The expansion of coverage will inevitably be accompanied by the low threshold of the pension system. We find that the larger the coverage rate is, the larger the pension gap will be. It is true that coverage expansion will be harmful to financial sustainability while advantageous to social protection. However, policy makers have another choice, using safety net (social assistance) to guarantee the most vulnerable elderly. Therefore the social insurance system could run independently without too much government intervention, just like what German pension system does. Anyway, we present possible results for policy makers who make reform decisions.
3. The simulations that are presented clearly present potential results of different policy options. In the comparison (part 5.4), as I understand, the results of all scenarios (but as separate reforms) are presented. It would be interesting to see what happens if all proposed measures are introduced in Urban Pension System, as some of the measures may interact, increasing or decreasing the total impact of the changes.
This is a very important suggestion. In the revised version, we have conducted scenario analysis on combination of different policy parameters (see 5.3).Thank you very much.
4. I have also found some editorial mistakes:
line 33, there should be plural form of the verb (point out)
line 93, there should be singular form for the verb (has)
line 120, the pension system does not have an age structure, it is rather the participants, in this particular context it is the comparison of contributors and beneficiaries.
All the mentioned mistakes have been revised and the language in the revised version have been checked. Thank you very much.

Reviewer 3 Report
The paper examines how the pension gap between the revenue and the expenditure in China changes in the future. As shown by an aging population in the economically developed countries, the population aging in china is the serious problem and the pension budget balance should be examined as the pension sustainability.
Therefore, this paper has some contributions as the research paper. This paper surveys the system of the pension in China and the reform in detail. However, I have some questions about the contribution for the publication.
This paper examines the balance of the pension numerically, based on the parameters considered by the authors. I am afraid whether the results obtained by this paper have the robustness or not. This paper considers the many changes of economic situations such as population aging, wage rate and others. However, the parameter settings show the one of many cases that we can consider.
Moreover, the pension system will be reformed in the future. For instance, the pension system in Japan was changed at the many times. The pension system is politically changed. Not to say, it is difficult to consider the political change at the numerical simulation.
In addition, the paper should consider general equilibrium. If the unemployment rate and the contribution rate change, the labor demand and the wage rate change. Moreover, the capital accumulation can be changed and then interest rate changes. This paper does not consider the effect of general equilibrium in detail.
The authors check the related literatures. However, I think that the authors should survey the related literatures such as the generational accounting that is considered by Kotlikoff and others. The generational accounting considers the inter-generational burden and this burden attributes to the social security such as pension.
The author should consider the abovementioned comments for the publication.
Author Response
Comments & Responses:
1. This paper examines the balance of the pension numerically, based on the parameters considered by the authors. I am afraid whether the results obtained by this paper have the robustness or not. This paper considers the many changes of economic situations such as population aging, wage rate and others. However, the parameter settings show the one of many cases that we can consider.
This is a very good suggestion. In the revised version, we have introduced time series model into the simulation, including probabilistic error and confidence interval to make the prediction as robust as possible. Moreover, in the sensitivity analysis, we not only consider the change of one factor but also consider the combined effect of policy parameters in various scenarios which may probably happen in the future.
2. Moreover, the pension system will be reformed in the future. For instance, the pension system in Japan was changed at the many times. The pension system is politically changed. Not to say, it is difficult to consider the political change at the numerical simulation.
This is a very important point. In the sensitivity analysis, we then include demographic factors as well as policy parameters. It is true that the pension system could be politically changed, however, the future direction of reform is predictable in China. Nowadays, under the background of economic downturn and structural reform of supply side, it is predictable that reduction of tax and fees is one of the major reform measures in the next decades. Therefore the paper simulate pension gap under various policy parameters, presenting possible reform effects on future pension sustainability.
3. In addition, the paper should consider general equilibrium. If the unemployment rate and the contribution rate change, the labor demand and the wage rate change. Moreover, the capital accumulation can be changed and then interest rate changes. This paper does not consider the effect of general equilibrium in detail.
This is also an important issue. General equilibrium model considers internal relationship among unemployment rate, the contribution rate change, the labor demand and the wage rate. But unfortunately the authors are not very familiar with the model and it takes time to rearrange the research using GE model. This research focuses on predict future trend of pension sustainability under stochastic environment (setting time series model for the average wage growth rate). We are going to solve this problem in the next paper.
4. The authors check the related literatures. However, I think that the authors should survey the related literatures such as the generational accounting that is considered by Kotlikoff and others. The generational accounting considers the inter-generational burden and this burden attributes to the social security such as pension.
Yes, generational accounting in General Equilibrium is very important in pension sustainability analysis, but applying the Auerbach-Kotlikoff Dynamic Life-Cycle Simulation Model into our analysis need more time and we are going to adopt this model in the next research. Thank you very much for your great suggestions.
The author should consider the abovementioned comments for the publication.

Round 2
Reviewer 1 Report
Referee report about the paper "Evaluation on the Sustainability of Urban Public Pension System in China", submitted by Qing Zhao and Haijie Mi for publication in Sustainability.
The new version of the paper has improved with respect the previous version. But I think that the authors must answer to some comments below. The first was included in my first report, but it was not answered.
Comments:
- The topic of the paper about the sustainability of a pension system is very interesting nowadays. A model to forecast the pension gap is considered, but the contribution with respect other models in the literature is not explained. For instance, [20] and [22] analyze the sustainability of a pension system but the relation with the current paper must be explained. Only a sentence in the introduction is included. There exist other papers with similar models? On the other hand, the methodology used only seems suitable for one of the public pension systems in China. Can be extrapolated to pension systems of other countries?
- Completing previous comment, [23] deals about urban workers also. A brief comment about the aim of that paper and the contributions of the present paper with respect to it must be added. In fact, in my first report, I suggested a new section where the authors analyze with their model the implications of changes of the demographic and economic policies on the pension gap evolution in order to diminish the gap. Instead, the authors have included some lines at the end of the conclusions. If it is possible, some implications with numbers must be included, not necessarily as a new section. For instance, you can infer the pension gap if the number of births increases a percentage.
Minor comments:
- Replace “[20-21].In” by “[20-21]. In” in line 87.
- Explain and indicate if Table 1 is obtained from [23] and [24], in line 222.
- If there exist a reference for equation of line 328 and for Leslie matrix, it should be included. Replace “ Where: P(t)” by “ where P(t)” in line 329, and substitute “;” by “,” (or “and”) in lines 329 and 330. Same comments after equations at lines 380, 389 and 394.
- Explain what is w in line 335.
- Figure 3 goes to 2100. According to the rest of the paper, it only should include the forecast to 2070.
- Remove “; AveWage(t-1) represents the average wage of urban employees in the year of t-1.”, in lines 391-392 because has been described previously.
- Equations (1), (2), …, (10) are not cited: remove the numbers.
- Why the pension gap is smaller in this new version of the paper? It is 3.36 trillion yuan instead 50 trillion yuan (approximately). See, for instance, previous comments to Figures 5-7.
- Replace [17][37] by [17] and [37], in line 424, and [11][20] by [11] and [20], in line 426, and
Wang and Rong (2012) by [34], in line 429, and Feng (2013) by [40], in lines 493-494.
- Table 4 must be cited in the text, Section 5.
- Revise tabulation previous to references 21, 23, 45 and 46.
Author Response
Comments & Responses
- The topic of the paper about the sustainability of a pension system is very interesting nowadays. A model to forecast the pension gap is considered, but the contribution with respect other models in the literature is not explained. For instance, [20] and [22] analyze the sustainability of a pension system but the relation with the current paper must be explained. Only a sentence in the introduction is included. There exist other papers with similar models? On the other hand, the methodology used only seems suitable for one of the public pension systems in China. Can be extrapolated to pension systems of other countries?
Thank you for your suggestion. Perhaps I have not clearly responded earlier. The basic model for calculating flow-based pension gap is first proposed and developed by the Office of the Chief Actuary of US. Social Security Administration. There are three basic types combining demographic, economic and pension factors to calculate pension revenues and pension expenditures: deterministic model, stochastic model and micro-simulation model (see line 58 to 84). These are the general models for other countries to follow although the specific model settings (like those presented in Section 4.2.2) would vary from country to country because of different policies in different countries. In China, when building the flow-based pension gap model for the public urban pension system, different researchers use different methods and assumptions. For example, [22] uses deterministic models (with the weakness that the prediction is not robust), and [20] uses stochastic models without considering important (combined) policy parameters. I have illustrated in line 86 to 96. However, in this paper we use stochastic models with particular attention on (combined) policy changes to make the prediction robust and close to reality. This is probably our contribution compared with previous research.
There is another thing we need to clarify. The reason why we only focus on the public pension system for urban employees in China is that it is the traditional Pay-As-You-Go Bismarck model with contributions from employers and employees, which should be quite independent from public finance. In this sense, research on pension sustainability is meaningful. However, the public pension system for urban and rural residents is more like a “zero-pillar” pension system dependent on enormous government subsides, therefore we seldom talk about the financial sustainability of this system.
- Completing previous comment, [23] deals about urban workers also. A brief comment about the aim of that paper and the contributions of the present paper with respect to it must be added. In fact, in my first report, I suggested a new section where the authors analyze with their model the implications of changes of the demographic and economic policies on the pension gap evolution in order to diminish the gap. Instead, the authors have included some lines at the end of the conclusions. If it is possible, some implications with numbers must be included, not necessarily as a new section. For instance, you can infer the pension gap if the number of births increases a percentage.
The change of economic factors has been considered by establishing the time series model of wage growth rates. Demographic factor of mortality has also been discussed in section 5. For fertility rate, as we have explained in 268 to 275, is quite uncertain in China now, and many papers have already discussed this, like [20]. In this paper, we mainly focus on the effect of possible combined policy changes on pension gap, in order to provide some policy implications for future reforms, which is also the distinguishing goal of this research compared with others.
Minor comments:
Thank you so much for your careful inspection. Most of the mistakes have been revised.
- Replace “[20-21].In” by “[20-21]. In” in line 87.
- Explain and indicate if Table 1 is obtained from [23] and [24], in line 222.
- If there exist a reference for equation of line 328 and for Leslie matrix, it should be included. Replace “ Where: P(t)” by “ where P(t)” in line 329, and substitute “;” by “,” (or “and”) in lines 329 and 330. Same comments after equations at lines 380, 389 and 394.
- Explain what is w in line 335. Age limit
- Figure 3 goes to 2100. According to the rest of the paper, it only should include the forecast to 2070.
- Remove “; AveWage(t-1) represents the average wage of urban employees in the year of t-1.”, in lines 391-392 because has been described previously.
- Equations (1), (2), …, (10) are not cited: remove the numbers.
- Why the pension gap is smaller in this new version of the paper? It is 3.36 trillion yuan instead 50 trillion yuan (approximately). See, for instance, previous comments to Figures 5-7.
This is because the assumption of the wage growth rate has changed from deterministic environment to stochastic environment.
- Replace [17][37] by [17] and [37], in line 424, and [11][20] by [11] and [20], in line 426, and
Wang and Ren (2012) by [34], in line 429, and Feng (2013) by [40], in lines 493-494.
- Table 4 must be cited in the text, Section 5.
It has been cited in line 534.
- Revise tabulation previous to references 21, 23, 45 and 46.
Reviewer 3 Report
Based on the reviewer comments, the authors reply appropriately.
Some parts of comments are considered in the revised paper. However, some parts of comments are not considered in this revised paper and the authors examine as the future research.
I think that it is examined in this paper not as the future research. However, because of data availability and model setting modification, I got it that the revision paper contained the effort of the authors and the reminder of the comments are considered as the future research.
Thank you for giving me the reviewer opportunity.
Author Response
Thank you so much for your good suggestion!